# Effect of Feller-Buncher Model, Slope Class and Cutting Area on the Productivity and Costs of Whole Tree Harvesting in Brazilian Eucalyptus Stands

**Ricardo Hideaki Miyajima [1], Paulo Torres Fenner [1], Gislaine Cristina Batistela [2]** and **Danilo Simões [2,\*]**

[1] Department of Forest Science, Soil and Environment, School of Agriculture, São Paulo State University (Unesp), Botucatu 18610-307, Brazil; miyajimaricardo@gmail.com (R.H.M.); paulo.fenner@unesp.br (P.T.F.)

[2] Production Engineering Coordination, Campus of Itapeva, São Paulo State University (Unesp), Itapeva 18409-010, Brazil; gislaine.batistela@unesp.br

\* Correspondence: danilo.simoes@unesp.br; Tel.: +55-015-3524-9100

**Abstract:** The operational productivity and costs of tree felling operations can be influenced by several factors, among which, the machine characteristics, slope class, the cutting area and the individual volume of the trees stand out. Thus, the main objective of the present study was to analyze the productivity and production cost for two feller-bunchers with different technical characteristics operating in a eucalyptus forest. The productivity was calculated from a time study and the factors analyzed were two feller-buncher models, two slope classes, and two cutting areas. The machine cost per scheduled hour was based on the methodology of the Food and Agriculture Organization of the United Nations. Analysis of the results showed that the felling and turn operational elements occupied the most time in the operational cycle of feller-bunchers. The machine cost per scheduled hour was USD 69.69 $h^{-1}$ for feller-buncher 1 and USD 102.03 $h^{-1}$ for feller-buncher 2. In conclusion, the distinct technical characteristics of feller-bunchers were found to influence the productivity and, consequently, the cost, of the felling operation during the harvesting of whole eucalyptus trees.

**Keywords:** forest operations; mechanized harvesting; performance measurement; operational planning; economic comparison





## 1. Introduction

In Brazil, the area occupied by commercial plantations represents 9.5 million hectares. Among the planted trees, eucalyptus accounts for 7.3 million hectares, with an average productivity of 35 m³/ha/year. The main uses of eucalyptus are pulp and paper, wood panels, laminated floors, charcoal, and solid wood products [1,2].

Among the positive aspects of the use of eucalyptus is the rotation time, which averages six years, and is shorter than that of other forest species planted in Brazil. The eucalyptus rotation time is related to the synergy of genetic improvement, forestry characteristics, favorable climatic conditions, and mechanization in the wood harvesting process. Due to their importance [3–7], eucalyptus plantations have been evaluated in terms of different aspects related to mechanized wood harvesting, using a diverse range of machines.

The scientific knowledge of technical-economic factors is essential for the decision-making process in the mechanized wood harvesting field. A clear understanding of this information can allow managers to make more plausible choices of harvest modes, particularly with a view to operational optimization and, therefore, minimizing the costs of the felling operation. Thus, determining productivity is of utmost importance. However, monitoring productivity may be complex because it can be influenced by several variables.

Previous research has aimed to derive accurate estimates for the cost of felling operations to effectively manage wood harvesting [8–12]. Therefore, the evaluation of productivity and costs involved in the cutting stage is important, because this enables improvements

in the production process and the quality of work, and greater competitiveness between companies. Reflecting the importance of this stage in the production chain, the feller-buncher is among the commercially available harvesting machines.

The feller-buncher comprises tire wheelsets or tracks, a cutting head, and a hydraulic device that cuts down the tree as it comes closer to the ground, accumulates a certain number of trees, then deposits them on the ground, forming wooden bundles that are suitable for extraction [13–17]. The felling of trees can be affected by factors such as slope, soil type, the machine used, forest type, volume per tree, spacing, forestry treatments, working system, operator, timber purpose, sensitivity of areas, priority order, silvicultural prescription, and cutting area [18–25].

Therefore, identifying the factors that may interfere with the productivity—and, consequently, the cost—of the felling operation is crucial, because these wood harvesting costs represent a considerable share of the final product cost [26–29], and are essential for achieving economic balance within the operation [30,31]. Although different techniques exist for estimating costs in timber harvesting, each has its own disadvantages [32,33].

A further potential complication is that costs are divided into capital costs and operating costs. Thus, the productivity of forest machines and the respective costs for wood harvesting operations are related to the planning of activities and their optimization [34–39].

Therefore, it is hypothesized that two feller-buncher models with different technical characteristics may influence productivity, machine costs per scheduled hour, and, subsequently, the felling operation cost. From this perspective, the objective was to evaluate the influence of operational conditions, considering two different feller-bunchers, two classes of slope, and two cutting areas, on productivity and production cost, based on experimental data related to eucalyptus felling.

## 2. Materials and Methods

### 2.1. Study Area

The study was conducted in the state of São Paulo, Brazil (22°84′25″ S, 48°34′19″ W). In this area, the average relative humidity is approximately 73%, with average annual rainfall of 1358 mm, and a dry season from June to September. According to the Köppen–Geiger classification, the climatic characteristics of the region are characterized as a Cwa tropical altitude climate [40], with rainfall in the summer and drought in the winter, and the warmest month average temperature over 22 °C.

The soil types of the harvest unit are RQo2 Neossol Arctic Quatzarenic Argissolic, A moderate or weak, and dystrophic [41]. The considered slope classes in the study are classified to a single field, and are separated according to the Brazilian soil classification [36]; that is, the slope classes (SC) are characterized as: Class 1: slope 0 to 3% (flat); Class 2: slopes 3.1 to 8% (smooth undulating).

For this study, a 72 month old first rotation eucalyptus plantation was selected, initially planted at a spacing of 3 × 2 m, with a density of 1638 trees per hectare at the time of harvest. The plantation was intended for the manufacture of wood panels and laminate floors. The trees had an average diameter at breast height (DBH) of 14.95 cm ± 3.62 cm and an average height of 20.87 m ± 3.28 m.

The individual tree volume was calculated using the model proposed by Schumacher and Hall [42] for forest inventory, applied to randomized plots; thus, we adopted plots with a size of 400 m$^2$. As a result, a total of 10 diameter classes were obtained, and volumetric calculations of five trees were then carried out for each diametric class [43].

The above calculations resulted in an individual average volume of 0.20 m$^3$ (merchantable timber with bark). The value of the individual average volume and the number of trees per cycle of machine were used in the calculation of productivity.

### 2.2. Data Collection

The whole tree harvesting was carried out with two feller-buncher models, which were both operated by a single operator. The details of the models are as follows:

Feller-buncher 1 (FB 1, Figure 1a): manufactured by Caterpillar, model 320 D FM; 5197 cumulative hours of use; 117 kW rated power motor; tracked undercarriage; crane with maximum reach of 5.90 m and Quadco model cutting head, with a cutting capacity of 457 mm and accumulation capacity of 0.28 m$^2$; and new purchase price of USD 301,057.

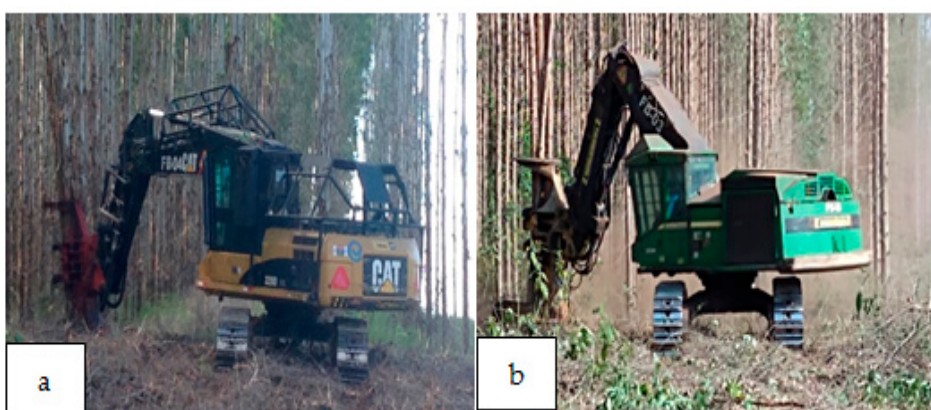

**Figure 1.** (**a**): Feller-buncher, Caterpillar. (**b**): Feller-buncher, John Deere.

Feller-buncher 2 (FB 2, Figure 1b): John Deere model 903 k; 13,241 cumulative hours of use; 224 kW rated power motor; tracked undercarriage; crane with maximum reach of 6.71 m and FS22B model forest implement, with a 558 mm cutting capacity and 0.48 m$^2$ build capacity. The new purchase price of the machine was USD 395,180.

The cutting areas (CAs) of the feller-bunchers were: CA 1—three rows of trees, i.e., nine meters wide; and CA 2—four rows of trees, i.e., 12 m wide. Both feller-bunchers performed in the cutting area. The tree bundles were positioned on the ground at an angle of 90°. The study area evaluated for FB 1 was 3.33 ha and for FB 2 was 3.50 ha. The two cutting areas (CA 1 and CA 2) were in the same location.

For assessment of the performance in the forest, a time and motion study was undertaken using a sexagesimal digital stopwatch, without stopping the time. The feller-buncher operating cycle consisted of three elements: tree felling (felling and moving), which started with the felling of the trees and ended when the felling head reached its accumulation capacity; tree bundle shifting (bunching)—the time required for deposition of the wood bundles on the ground; and return to felling (positioning) after finishing the bunching—the time taken to position the felling head on the tree to be felled.

The treatments were characterized as follows: T1 = (FB 1, SC 1, CA 1); T2 = (FB 1, SC 1, CA 2); T3 = (FB 1, SC 2, CA 1); T4 = (FB 1, SC 2, CA 2); T5 = (FB 2, SC 1, CA 1); T6 = (FB 2, SC 1, CA 2); T7 = (FB 2, SC 2, CA 1); T8 = (FB 2, SC 2, CA 2); (Figure 2).

The average time taken by delays or interruptions was 3.98 h, corresponding to 20.8% of the machine's programmed use time. These delays related to the time used for machine repairs and maintenance, and the physiological needs of the operator. For each operational cycle, the number of trees was counted (Table 1).

A total of 120 operating cycles were observed (pilot study) to determine the sample size, i.e., to determine the maximum allowed difference between the population mean μ and the sample mean $\bar{x}$, according to Equation (1), with a confidence level of 95% and error of 5%.

$$n = \left( \frac{z_g \, s}{e} \right)^2 \tag{1}$$

| $n$ | is the sample size; |
| $z_g$ | is the abscissa of the standard normal distribution, with a g level of confidence; |
| $s$ | is the standard deviation of the sample; |
| $e$ | is the allowable error. |

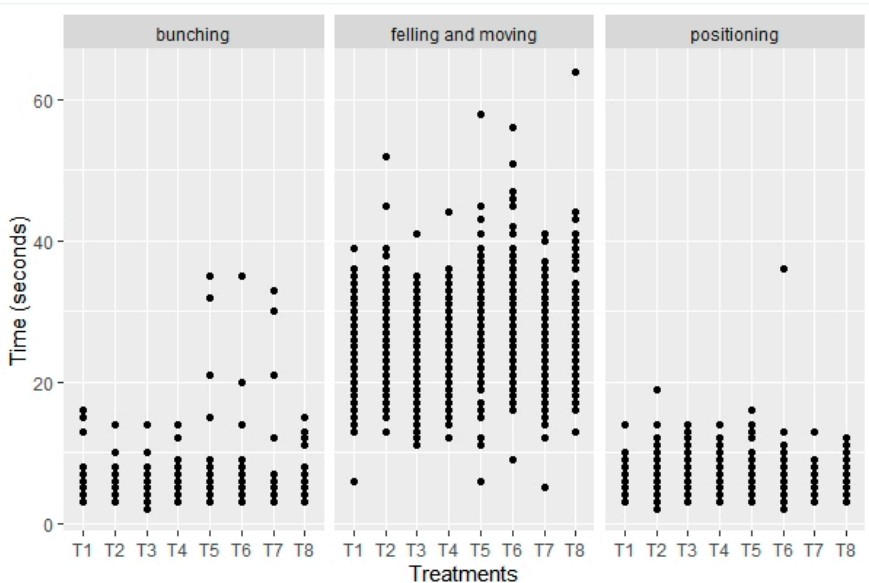

**Figure 2.** Time elements for each treatment.

**Table 1.** Descriptive statistics (number of cycles, minimum, maximum, average) for the number of trees felled by the feller-bunchers.

| FB | SC | CA | |
|---|---|---|---|
| | | **1** | **2** |
| 1 | 1 | (236; 3; 8; 5.390) | (308; 3;9; 5.484) |
| | 2 | (218; 3; 8; 5.509) | (249; 4; 8; 5.582) |
| 2 | 1 | (205; 2; 11; 7.190) | (214; 3; 10; 7.584) |
| | 2 | (185; 2; 10; 6.735) | (222; 4; 10; 6.784) |

Subtitle: FB: 1 (feller-buncher 1); 2 (feller-buncher 2); SC (slope class): 1 (flat); 2 (mildly undulating); CA (cutting area): 1 (site composed of three rows); 2 (site composed of four rows).

Productivity reflects the relationship between the volume of wood felled (number of trees per cycle × individual average volume, which was obtained from the rigorous cubing of 50 trees) by the forest machine in cubic meters, and the productive machine hour (without delays or interruptions). The productivity was calculated according to Equation (2).

$$P = \frac{VOL}{PMH} \tag{2}$$

$P$ is the productivity (m³ h⁻¹);
$VOL$ is the volume of wood felled (m³);
$PMH$ is the productive machine hour (h).

*2.3. Economic Analysis*

Because the US dollar is considered an international reference currency, monetary values were expressed in USD, according to official value of the currency at the official selling price of the Brazilian Central Bank. Therefore, the foreign currency price measured in units and fractions of the national currency was considered to be the exchange rate, which was BRL 3.31 on 3 April 2019, according to data provided by the Central Bank of Brazil [44].

The machine costs per scheduled hour were estimated using an accounting method, in line with the cost control methodology for timber harvesting, recommended by the Food and Agriculture Organization of the United Nations [45]. In this method, depreciation costs, the opportunity cost of capital, labor, insurance, and property taxes were classified as

fixed. The variable costs included monetary values of expenditures on fuel, maintenance and repairs, spare parts, and lubricating oils and greases.

The rate of the opportunity cost of capital relating to the acquisition of the machines was calculated using the Weighted Average Cost of Capital (WACC). This reflected the share of third-party capital in the company considered in this study. The WACC is the discount rate that comprises a weighted average of the marginal cost of capital, after discounting taxes, of each source of capital (Equation (3)).

$$WACC = c_c(1 - \iota\tau)\frac{H}{H + S} + c_s\frac{S}{H + S} \tag{3}$$

$c_c$   is the third-party capital cost;
$\iota\tau$   is the income tax rate;
$H$   is the present value of the debt holders' wealth;
$S$   is the present value of shareholders' wealth;
$c_s$   is the shareholders' capital cost.

Thus, $k_b$ was represented by the expected return adjusted for the risk involved in the operation. This value is equivalent to the fixed income return rate issued by the United States Treasury Department because it is considered to be a real and risk-free interest rate. It is also integrated into the global capital market, as advocated by Assaf Neto [46]. The Capital Asset Pricing Model [47] was applied to calculate the return required by shareholders, i.e., $k_s$. Moreover, a country risk premium was considered because the evaluation was undertaken in an emerging economy [48], as expressed by Equation (4).

$$r_p = r_f + \beta\left(r_m - r_f\right) + \lambda \tag{4}$$

$r_p$   is the expected return on the company's common equity;
$r_f$   is the risk-free rate of return—U. S. Treasury 10 Y bonds;
$\beta$   is the beta coefficient, which represents the project-specific risk factor—Brazilian companies in the forest products sector;
$r_m$   is the return rate of the market portfolio—S & P Global Timber & Forestry Index 10 year;
$\left(r_m - r_f\right)$   is the market risk premium;
$\lambda$   is the Brazil risk premium—Emerging Markets Bond Index Plus.

Thus, for more adequate planning of timber harvesting, the remuneration of third-party capital was 5.96%, the proportion of assets financed by debt was 40.84%, and the increased income tax rate was 34%. Considering the return on equity of 6.85%, the risk-free interest rate was 2.36%. The systematic risk coefficient of the asset was the average total beta, with an average unleveraged beta of 0.33. After re-leveraging, the re-leveraged beta was 0.42.

The annualized market portfolio return was 5.06% and the country risk premium of the Emerging Markets Bond Index (EMBI + Br) was 2.37%. Thus, the estimated opportunity cost of capital was 9.16%. The weighted values were based on geometric averages, as recommended by Damodaran [49].

In addition, the economic life of the machines was estimated to be five years, with a resale value of 20% of the purchase price [45]. The social charges and social benefits corresponded to 134% of the machine operator's salary, and this percentage was obtained from the forestry company.

The cost of the felling operation (CFO) is the ratio between the machine cost per scheduled hour and the productivity (Equation (5)):

$$CFO = \frac{MCH}{P} \tag{5}$$

*CFO* is the cost of the felling operation (USD m$^{-3}$).

*MCH* is the machine cost per scheduled hour (USD h$^{-1}$).

## 2.4. Statistical Design

In this experiment, three independent factors were considered with two levels each, described as follows: factor A refers to the model of feller bunchers, factor B refers to the slope class of the terrain, and factor C describes the cutting area. In this scenario, the combinations of the levels of the factors resulted in the treatments of the experiment, characterized as a $2^3$ factorial scheme arranged in a completely randomized design (CRD).

Thus, the element $y_{ijkl}$ represents the *l*-th repetition ($l = 1, 2, \ldots, n_{ijk}$, where $n_{ijk}$ is the number of observations per treatment) of the *i*-th level of factor A ($i = 1, 2$), the *j*-th level of factor B ($j = 1, 2$), and the *k*-th level of factor C ($k = 1, 2$). According to Montgomery [45], the general model of the response variable (Equation (6)) is:

$$y_{ijkl} = \mu + \tau_i + \beta_j + \gamma_k + (\tau\beta)_{ij} + (\tau\gamma)_{ik} + (\beta\gamma)_{jk} + (\tau\beta\gamma)_{ijk} + \epsilon_{ijkl} \tag{6}$$

$\mu$    is the overall mean effect;
$\tau_i$    is the effect of the *i*th level of the factor A;
$\beta_j$    is the effect of the *j*th level of the factor B;
$\gamma_k$    is the effect of the *k*th level of the factor C;
$(\tau\beta)_{ij}$ is the effect of the interaction between $\tau_i$ and $\beta_j$;
$(\tau\gamma)_{ik}$ is the effect of the interaction between $\tau_i$ and $\gamma_k$;
$(\beta\gamma)_{jk}$ is the effect of the interaction between $\beta_j$ and $\gamma_k$;
$(\tau\beta\gamma)_{ijk}$ is the effect of the interaction between $\tau_i$, $\beta_j$ and $\gamma_k$;
$\epsilon_{ijkl}$    is a random error component.

The operating cycle element times, the productivity, and the cost of the felling operation (per m$^3$) evaluated in these different treatments were compared using the three factor analysis of variance model (ANOVA), complemented by the Tukey–Kramer test for multiple comparisons, considering a 5% significance level [50]. Statistical analyses were performed using the Genmod procedure of the statistical software SAS [51].

Furthermore, for the collected data, using the Kolmogorov–Smirnov and Bartlett tests, the assumptions of normality and homogeneity of variances were not able to be verified. However, according to the Central Limit Theorem [52,53], the distribution of the sample averages tends to be normal as the sample size grows, regardless of the distribution of the variable of interest, for large samples (n > 30). Multiple regression modeling was carried out using R software version 3.5.2 [54].

## 3. Results

Considering the sample sufficiency for FB 1, 1011 operational cycles were observed, which resulted in the cutting of 1110.40 m$^3$ (total number of trees × individual average volume). For FB 2, 826 operational cycles were observed, which amounted to 1169.80 m$^3$ of wood. In addition, 10.02 working hours were observed for FB 1, with an accuracy level of 1.61%; and 9.11 working hours were observed for FB 2, with an accuracy level of 1.53% (which was calculated using Equation (1)).

The analysis of the influence of the operating area on the felling element (Table 2) showed a statistically significant difference ($p < 0.05$) for all feller-buncher operations. It was also found that the shortest time for the felling element to be performed was obtained at the location of the three felling rows.

For the felling element of the operating cycle, only one operating condition showed no statistically significant difference ($p > 0.05$) for the influence of the slope class. However, for Class 2 it was verified that the times associated with the felling element were shorter compared to those of Class 1. Regarding the feller-buncher type, a statistically significant difference ($p < 0.05$) was found for all operational conditions, in which the time for the felling element was shorter for FB 1.

**Table 2.** Average time (seconds) ± standard deviation of the elements and descriptive statistics (number of cycles, minimum, maximum) of tree felling carried out by the feller-bunchers.

| FB | SC | CA | Felling | Bunching | Turn |
|----|----|----|---------|----------|------|
| 1 | 1 | 1 | 23.619 ± 4.918 a A α (236; 6; 39) | 5.030 ± 1.534 a A α (236; 3; 16) | 6.280 ± 1.587 a A α (236; 3; 14) |
| | | 2 | 25.857 ± 4.646 b B α (308; 13; 52) | 4.701 ± 1.219 a A α (308; 3; 14) | 6.299 ± 1.785 a A α (308; 2; 19) |
| | 2 | 1 | 22.940 ± 4.807 a A α (218; 11; 41) | 5.165 ± 1.378 b A α (218; 2; 14) | 6.633 ± 1.697 a B α (218; 3; 14) |
| | | 2 | 24.217 ± 4.485 b A α (249; 12; 44) | 4.763 ± 1.291 a A α (249; 3; 14) | 6.494 ± 1.535 a A β (249; 3; 14) |
| 2 | 1 | 1 | 27.044 ± 5.597 a B β (205; 6; 58) | 5.298 ± 3.261 a A α (205; 3; 35) | 6.380 ± 1.958 a B α (205; 3; 16) |
| | | 2 | 29.794 ± 5.938 b B β (214; 9; 56) | 4.808 ± 2.561 b A α (214; 3; 35) | 6.089 ± 2.581 a A α (214; 2; 36) |
| | 2 | 1 | 24.065 ± 5.077 a A β (185; 5; 41) | 5.497 ± 3.107 a A α (185; 3; 33) | 5.816 ± 1.233 a A β (185; 3; 3) |
| | | 2 | 25.630 ± 5.432 b A β (222; 13; 64) | 5.248 ± 1.429 a B β (222; 3; 15) | 5.941 ± 1.386 a A α (222; 3; 12) |

Lower case letters compare the cutting area according to the Tukey–Kramer test with 5% significance. Upper case letters compare slope classes, according to the Tukey–Kramer test with 5% significance. Greek letters compare feller-bunchers according to the Tukey–Kramer test, with 5% significance. Subtitle: FB: 1 (feller-buncher 1); 2 (feller-buncher 2); SC (slope class): 1 (flat); 2 (mildly undulated); CA (cutting area): 1 (site composed by three rows); 2 (site composed by four rows).

Regarding the influence of the cutting area on the bunching element, it was found that in 50% of operational conditions there was a statistically significant difference ($p < 0.05$), in which the average time of this element was shorter when the work area consisted of four rows.

Concerning the slope effect, only one operating condition was statistically significantly different ($p < 0.05$). Moreover, the time required to perform the bunching element related to Class 1 was shorter than that for Class 2. Regarding the influence of the machine type on the time to perform the bunching element, a statistically significant difference was found for only one operational condition ($p < 0.05$). Under all conditions, a longer time for the performance of the bunching element was found for FB 2.

In addition, regarding the time to perform the positioning element, there was no statistically significant difference ($p > 0.05$). Regarding the slope class effect, 50% of the operating conditions presented statistically significant differences ($p < 0.05$). The FB 1 time for the performance of this element was longer than that for Class 2, whereas for FB 2, this was observed for Class 1.

Regarding the analysis of the influence of the feller-buncher, there were statistically significant differences ($p < 0.05$) in 50% of the operational conditions. This may indicate the influence of the machine type on the positioning element, and that the situations in which they occurred were associated with Class 2, which required a shorter time for FB 2.

Among the technical information involved in the mechanized harvesting of wood, information about feller-buncher productivity is essential because it can influence the decisions and planning of the forest manager.

Figure 3 shows the box plots for productivity in each treatment. The visual inspection of these box plots indicates the presence of outliers identified in all analyses. The plots also show that the medians of productivity are higher for FB 2 than FB1, regardless of the levels of SC and CA.

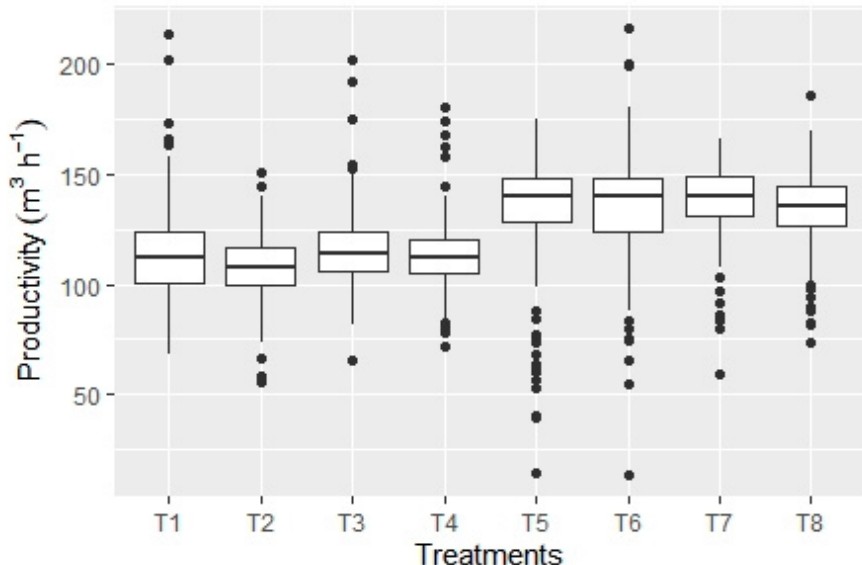

**Figure 3.** Box plots of productivity per treatment.

The analysis of variance for productivity indicates that the interaction between the three factors is statistically significant ($p = 0.002$), i.e., factor A refers to the two models of feller-bunchers, factor B refers to the slope classes of the terrain, and factor C describes the cutting area.

The analysis of the effect of the cutting area (CA 1 and CA 2) on the productivity (Table 3) showed there was no statistically significant difference ($p > 0.05$) in the operating conditions. However, in these operational conditions, it was verified that the cutting area composed of three lines presented a shorter effective time for the performance of the operational cycle and a shorter time for the performance of the felling element.

**Table 3.** Average productivity $\pm$ standard deviation ($m^3 h^{-1}$) and descriptive statistics (number of records, minimum, maximum).

| FB | SC | CA | |
|----|----|----|----|
| | | **1** | **2** |
| 1 | 1 | 112.775 $\pm$ 20.054 a A $\alpha$ (236; 67.924; 213.333) | 107.834 $\pm$ 14.314 a B $\alpha$ (308; 55.385; 150.698) |
| | 2 | 114.805 $\pm$ 17.746 a A $\alpha$ (218; 65.455; 201.600) | 113.840 $\pm$ 15.371 a A $\alpha$ (249; 72.000; 180.000) |
| 2 | 1 | 133.897 $\pm$24.956 a B $\beta$ (205; 14.845; 174.546) | 136.026 $\pm$14.424 a A $\beta$ (214; 13.884; 216.000) |
| | 2 | 137.436 $\pm$ 18.217 a A $\beta$ (185; 59.016; 166.154) | 132.603 $\pm$ 17.311 a A $\beta$ (222; 73.469; 185.807) |

Lower-case letters compare the cutting area according to the Tukey–Kramer test with 5% significance. Upper-case letters compare slope classes according to the Tukey–Kramer test with 5% significance. Greek letters compare feller-bunchers according to the Tukey–Kramer test with 5% significance. Subtitle: FB: 1 (feller-buncher 1); 2 (feller-buncher 2); SC: slope class, 1 (flat); 2 (mildly undulated); CA: cutting area, 1 (site composed by three rows); 2 (site composed by four rows).

The analysis of the slope class effect showed that a statistically significant difference ($p < 0.05$) was found for the feller-buncher FB1 in the cutting area CA 2, and for the feller-buncher FB 2 in the cutting area CA 1, under the operating conditions. In the comparison between FB 1 and FB 2, statistically significant differences were found ($p < 0.05$) concerning productivity between the machines, operated by the same operator.

Figure 4 shows the box plots for the cost of the felling operation in each treatment. The visual inspection of the box plots shows the presence of outliers and that the medians

of the cost of the felling operation are higher for FB 2 than those of FB1, regardless of the levels of SC and CA.

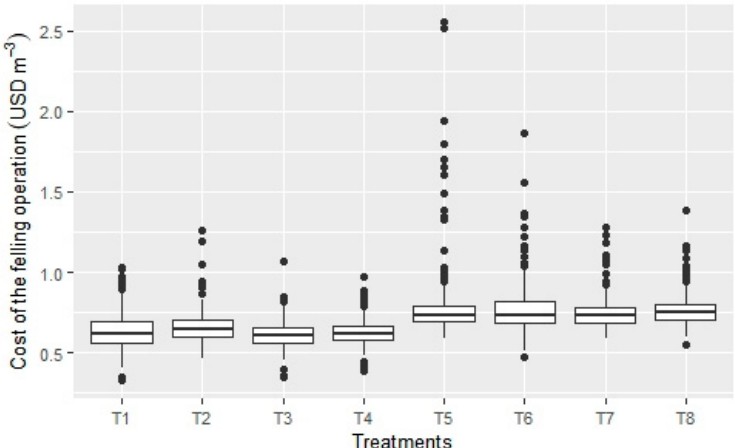

**Figure 4.** Box plot of the cost of the felling operation per treatment.

The analysis of variance for the cost of the felling operation showed that the interaction between the three factors is statistically significant ($p = 0.008$), i.e., factor A refers to the two models of feller-bunchers, factor B refers to the slope classes of the terrain, and factor C describes the cutting area.

Therefore, the machine cost per scheduled hour for FB 1 was USD 69.69 $h^{-1}$. Among the components, the fuel cost was the most significant, representing 32.40%, followed by labor cost, representing 19.73%, and depreciation, which represented 17.28%; that is, in total, these components represented 69.41% of the machine costs per scheduled hour.

For FB 2, the machine cost per scheduled hour was USD 102.03 $h^{-1}$. The fuel cost corresponded to 42.28%, depreciation corresponded to 15.49%, and the labor cost corresponded to 13.47%. These elements were the most significant in the composition of the machine costs per scheduled hour, in total accounting for about 71.24%. The analysis of the influence of the cutting area on the cost of the felling operations (Table 4) showed that no statistically significant difference ($p > 0.05$) was found for the evaluated operating conditions.

**Table 4.** Average ± standard deviation of the cost of the felling operation (USD $m^{-3}$).and descriptive statistics (number of records, minimum, maximum).

| FB | SC | CA | |
|----|----|----|----|
| | | **1** | **2** |
| 1 | 1 | 0.637 ± 0.113 a A α (236; 0.327; 1.026) | 0.659 ± 0.098 a B α (308; 0.462; 1.258) |
| | 2 | 0.621 ± 0.094 a A α (218; 0.346; 1.065) | 0.623 ± 0.083 a A α (249; 0.387; 0.968) |
| 2 | 1 | 0.799 ± 0.263 a B β (205; 0.585; 2.551) | 0.774 ± 0.171 a A β (214; 0.472; 1.860) |
| | 2 | 0.753 ± 0.117 a A β (185; 0.585; 1.275) | 0.778 ± 0.115 a A β (222; 0.549; 1.389) |

Lower-case letters compare the cutting area, according to the Tukey–Kramer test with 5% significance. Upper-case letters compare slope classes, according to the Tukey–Kramer test with 5% significance. Greek letters compare the feller-bunchers, according to the Tukey–Kramer test with 5% significance. Subtitle: FB: feller-buncher, 1 (feller-buncher 1); 2 (feller-buncher 2); SC: slope class, 1 (flat); 2 (mildly undulated); CA: cutting area, 1 (site composed by three rows); 2 (site composed by four rows).

For the slope class, it was found that, in 50% of the operating conditions, there was a statistically significant difference ($p < 0.05$). The lowest cost of the felling operation was for Class 2. In addition, analysis of the influence of the feller-buncher showed that there were

statistically significant differences ($p < 0.05$) for all operating conditions, and the lowest cost of the felling operation was related to feller-buncher 1.

## 4. Discussion

Time studies are crucial for analyzing the time associated with the operational elements of the feller-buncher. They make it possible to identify loss of time in carrying out the operational elements and the factors that may contribute positively or negatively to the operation. These studies also provide information to improve productivity and reduce costs in felling operations.

The analysis showed that the influence of the cutting area, specifically for the site composed of three felling rows, was statistically significant ($p < 0.05$). A possible explanation is that, for this operating condition, the feller-buncher consumed less time in the felling and moving elements. Thus resulted in a shorter time required for felling and moving, compared to the cutting area consisting of four felling rows. Moreover, analysis of the influence of the feller-buncher model showed that a shorter time was associated with the felling and moving element for FB 1. This may indicate that the technical characteristics of the feller head and the rated power of the machine influence the time required for this element. Although the time required to perform the element was shorter, so also was the number of trees per cycle.

Although no significant differences ($p > 0.05$) were found for FB 2, in all operational conditions this machine required a longer time for the realization of the element. This may be explained by the higher number of trees per cycle for FB 2 and, therefore, the greater time required to perform the element. Analysis of the effect of the slope class and the feller-buncher model on the turning element showed there were significant differences in 50% of the operational conditions. This result may be attributed to the coefficient of variation, of 42%, which reflects the outliers present in the collected and analyzed data.

Analysis of the time required to perform the elements that make up the feller-buncher operational cycle in a eucalyptus forest showed that the greater the slope class, the longer the time taken for felling. Conversely, the greater the time for the felling element, the greater the slope class [6,55,56]. The time required in the current study to perform the felling element was greater than that found in the study of Diniz [57], for the same slope classes. A possible explanation for the shorter time for Class 2 may be related to the lower number of trees per cycle, which resulted in a shorter time associated with this element. In addition, bunching and positioning were the most time-consuming elements for the felling operation in a planted forest [58].

Among the technical information involved in wood harvesting, knowledge regarding productivity is essential. This information can be used to assist in forest manager decisions and planning. With the premise of analyzing the influence of different models of feller-bunchers in relation to productivity, statistically significant differences were found ($p < 0.05$); that is, technical differences in the machines were found to influence productivity. These results corroborate those of Acuna and Kellog [59] and Strandgard [4].

Analysis of the operating conditions in which there were statistically significant differences ($p < 0.05$) showed that the productivity in the cutting area composed of three rows was greater than that in the cutting area composed of four rows. These results differed from the results found for a work area composed of four rows and a wood bundle arrangement at 45° [60].

In other studies, various productivity values have been found, e.g., productivity of 40.84 m$^3$ h$^{-1}$ in the study of Fernandes [61] for a cutting area of four rows in a eucalyptus forest with flat slopes; and productivity of 109.1 m$^3$ h$^{-1}$ [62]. For a Pinus forest with a gently undulating slope, the average productivity was 103.8 m$^3$ h$^{-1}$ [63] and the maximum productivity was 117.7 m$^3$ h$^{-1}$ [64]. By comparison, for a eucalyptus forest, with a flat terrain and a cutting area composed of five rows, the productivity was found to be 48.80 m$^3$ h$^{-1}$ [65], and for a cutting area composed of three rows, the productivity was

found to be 127.50 m$^3$ h$^{-1}$ [66]. In the present study, the characteristics of the head and the rated power of the machine were able to explain these differences.

The slope class effect presented statistically significant differences ($p < 0.05$) for two operational conditions. This result can be explained by the fact that the effective time for felling and movement was greater for Class 1. However, under other operational conditions, the slope class did not show significant differences. This can be explained by the similarity of the slopes of the land, even though they were classified to different slope classes according to the Soil Classification. Although the slope class variable did not influence all of the evaluated operational conditions, slope is one of the factors that most influences the feller-buncher productivity [67–69].

Comparison of FB 1 and FB 2 showed statistically significant differences ($p < 0.05$) in productivity. This result did not corroborate that of Seixas and Batista [70]. These differences in productivity can be explained by the higher average number of trees per operating cycle for FB 2, because this machine was able to accumulate an average of 7.08 trees per cycle, whereas FB 1 was able to accumulate an average of 5.49 trees per operating cycle. Other influences include the number of trees and, according to other research, the wood volume, basal area, and species [71–76], and the average diameter at breast height [77,78].

The knowledge of the costs related to forest machines is fundamental for effective planning of harvesting operations. Information regarding the machine costs per scheduled hour allows the characteristics of machines to be used in the wood harvesting system to be chosen. Given this context, analysis undertaken in the current study showed that the machine costs per scheduled hour of FB 1 were less than USD 82.42 h$^{-1}$, which was determined for the cutting operation with a feller-buncher in planted forests [79], and for eucalyptus planted forest conditions with a flat terrain [80]. For FB 2, the machine costs per scheduled hour were higher than those of the values determined by Long and Wang [81]. These differences can be explained by the methodology adopted for each situation, in addition to the time and the rate used for machine depreciation, the monetary sums spent on labor and the respective social charges—which are inherent to each forest-based company—and the fuel expenses, which vary by region and country.

Therefore, the distinct machine costs per scheduled hour for the two feller-bunchers influence the cost of the felling operations, i.e., for FB 1, these costs were lower than those of FB 2 for all operating conditions. This difference in production cost can be attributed to the lower machine cost per scheduled hour of FB 1, which corroborates the fact that the cost of the felling operation has a direct relationship with productivity [63].

## 5. Conclusions

In conclusion, among the operational elements of the feller-bunchers, the felling, moving, and bunching elements required the most time to be carried out.

The findings obtained in a eucalyptus planted forest indicated that, among the analyzed factors, the differences in the technical characteristics of the feller-bunchers and the slope class influenced the productivity and production costs. Although the productivity of FB 1 was 16.8% lower than that of FB 2, the operational costs were 18% lower per m$^3$ due to the lower machine costs per scheduled hour of FB 1.

**Author Contributions:** Investigation: R.H.M.; Conceptualization: P.T.F.; Formal Analysis: G.C.B.; Methodology, Validation and Writing—Review & Editing: R.H.M. and D.S.; Supervision: D.S. All authors have read and agreed to the published version of the manuscript.

**Funding:** This work was carried out with the support of the Higher Education Personnel Improvement Coordination-Brazil (CAPES)-Financing Code 001.

**Conflicts of Interest:** The authors declare no conflict of interest.

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
