# Peer review of "Effect of Feller-Buncher Model, Slope Class and Cutting Area on the Productivity and Costs of Whole Tree Harvesting in Brazilian Eucalyptus Stands"

_forests, doi:10.3390/f12081092_

Round 1
Reviewer 1 Report
The paper needs a major revision. I have provided several and detailed comments in the pdf file attached.
The description of the statistical methods is good, but the description of the harvesting study is rather weak. For example, the literature review only presents a few previous studies in eucalyptus stands/operations, the description of the time elements is not clear, there is no info about bunch size and distances between bunches, the economic analysis needs to be redone applying more standards methods used in harvesting studies, the slope classes are not quite different, it is not clear the impact of operator experience on the results, etc. The Discussion of the results is also weak, and the Conclusion section needs to be improved.
Finally, some English typos must be corrected.

Author Response
"Please see the attachment"

Reviewer 2 Report
Although the aims of the research are useful for forest practitioners and the study is well designed, some aspects must be clarified and some results must be refined in order to have more exact and accurate predictive models than those developed by the authors.
Regarding the material and methods chapter, I consider better to use the Kwh as power unit rather than hp, I suggest to change ub¡nits in chapter 2.2. in the same chapter, I would like to be clarified about wether the purchasing prices of the machines refer to new machines or otherwise which were the machines' lifes at the purchasing moment. To me it is strange that a machine that more than doubles the power of the other were only 31% more expensive.
Behind the Figure 1, the phrase "For the delays or interrupptions the average time was 3.98 hours" is not clear enough, the authors should explain better (e.g. expressing the delay time as a percentage of the scheduled time) or rephase the sentence in order to clarify its meaning.
Regarding the equation (1), I think that the proper expression would be n=(tα/2·šn-1/e)2, being t the value of Student's t distribution and šn-1 the sample standard deviation. Although for n>100, student t approaches a normal distribution, the proposed expression is more general when one does not know the variance of the population, but only the estimation from a pilot sample.
Above the table 4, ther is a typo in "(CA1) operating conditionS", The authors should change the capital S by s.
Also in the Results chapter, the main problem of the analysis is very clear in Figure 3 and Figure 4, that show a strongly heterocedastic residuals distribution. In Figure 3, is clear that the model strongly overestimates the lower observed productivities while it strongly underestimates its greater values. the same fact is shown in Figure 4 regarding cost.
I think that the multiple linear regression employed by the authors is not good enough to estimate the actual productivity and costs. I suggest trying different models for each of the machines, using alometric transformations for the continuous variable (number of trrees per cycle) or trying other transformations to reduce heterocedasticity, such as Box-cox transformations.
In the present form, the predictive power of the model is very poor for the lower and greater values of the observed variables, so it must be corrected. As the object of the research is relevant enough, and the database is significant, I am sure that a refined statistical analysis would improve a lot their predictive models and would permit more significant conclusions.
Reviewer 3 Report
The paper is in accordance with my field of research and an interesting topic and I therefore accepted to review this manuscript. I found the paper very fluid, well structured with explicit aims. I suggest accepting the manuscript after minors revisions, please find below all the comments :
I suggest using whole-tree harvesting instead of full tree harvesting along the manuscript
Introduction
Please introduce Brazilian Eucalyptus stands (overview: %, species, wood uses, rotations…etc)
‘’The objective is to find accurate estimates for the cost of the felling operation in order to effectively manage the wood harvesting operations’’ …The aim of your study should be at the end of the introduction section (state the logical form of your problematic from the general (research theme) to the particular (hypothesis), from the abstract to the concrete… funnel principle
Materials and Methods
Study area : plantations density ? what purposes biomass-energy/timber ?
Data collection : I suggest adding two pictures for the feller-buncher machines
What about operators ? was it the same for the two machines ? please add details
Economic analysis
‘’the economic life of the machines was estimated at five years, with a resale value of 20% of the purchase price.’’ I think that after 5 years 20% is very optimistic ! please cite the reference for this estimation
Statistical design
factorial scheme 23 ?
Results
Move/merge with the text below the table
‘’ For the interpretation of Table 2, lower case letters compare the effect of CA, fixed FB and SC, upper case letters compare the effect of SC, fixed FB and CA and greek letters compare the effect of FB, fixed SC and CA, for each operational element. In the analysis of the influence of the operating area for the realization of the felling element (Table 2), a statistically significant difference was found (p<0.05) for all feller-buncher operating situ-ations. It was also noticed that the shortest time for the felling element to be performed was to the site of three lines. ‘’
Round 2
Reviewer 2 Report
Thank you for performing the requested changes, in its present form the article has gained significance.